# When Dynamic Data Selection Meets Data Augmentation: Achieving Enhanced Training Acceleration

**Suorong Yang**[*][1][2]  **Peng Ye**[2][3]  **Furao Shen**[1]  **Dongzhan Zhou**[2]

## Abstract

Dynamic data selection aims to accelerate training with lossless performance. However, reducing training data inherently limits data diversity, potentially hindering generalization. While data augmentation is widely used to enhance diversity, it is typically not optimized in conjunction with selection. As a result, directly combining these techniques fails to fully exploit their synergies. To tackle the challenge, we propose a novel online data training framework that, for the first time, unifies dynamic data selection and augmentation, achieving both training efficiency and enhanced performance. Our method estimates each sample's joint distribution of local density and multimodal semantic consistency, allowing for the targeted selection of augmentation-suitable samples while suppressing the inclusion of noisy or ambiguous data. This enables a more significant reduction in dataset size without sacrificing model generalization. Experimental results demonstrate that our method outperforms existing state-of-the-art approaches on various benchmark datasets and architectures, e.g., reducing 50% training costs on ImageNet-1k with lossless performance. Furthermore, our approach enhances noise resistance and improves model robustness, reinforcing its practical utility in real-world scenarios.

## 1. Introduction

Deep learning has thrived with the growing availability of large-scale datasets. As models become more complex and parameter-extensive, it is necessary to utilize even larger datasets, introducing challenges like reduced training efficiency. Moreover, large-scale datasets often include redundant or noisy samples (Xia et al., 2023b; Northcutt et al., 2021; Wang et al., 2018), which can compromise training effectiveness. To address these challenges, various data selection strategies have been proposed to reduce dataset size and enhance data efficiency while maintaining model performance. These methods can be broadly categorized into static data selection (Tan et al., 2024; Zhang et al., 2024; Xia et al., 2023b) and dynamic data selection (or pruning) (Qin et al., 2024; Raju et al., 2021; He et al., 2024). Static selection identifies a fixed subset of data before training begins, whereas dynamic pruning continuously selects the most influential samples during training. While these methods can effectively reduce training costs without degrading performance, the reduced training data volume often leads to reduced data diversity. Consequently, the generalization of deep models is limited, and lossless performance is typically achieved with relatively high selection ratios.

In fact, to address the issue of data diversity, data augmentation is commonly used in model training, which can also improve generalization (Xu et al., 2023; Yang et al., 2022b). However, effectively combining data selection and augmentation remains a challenge. Existing data selection methods typically prioritize representative and challenging samples, which are not specifically designed with augmentation in mind. Although augmentation can increase the diversity of selected data and further improve model robustness, applying it to complex samples may introduce ambiguity or noise, potentially increasing training difficulty (Gong et al., 2021; Yang et al., 2024b). Therefore, integrating data selection and augmentation in a unified framework presents a promising yet underexplored direction to balance efficiency and generalization.

In this study, we propose a novel framework that integrates dynamic data selection with augmentation, enabling a unified approach to enhance training efficiency and model generalization. During model training, low-density samples often correspond to underlearned or insufficiently represented data points, such as classification boundaries. Applying augmentation transformations to these samples reinforces

---

[*]This work was done during his internship at Shanghai Artificial Intelligence Laboratory. [1]National Key Laboratory for Novel Software Technology, Nanjing University [2]Shanghai Artificial Intelligence Laboratory [3]The Chinese University of Hong Kong. Correspondence to: Furao Shen <frshen@nju.edu.cn>, Dongzhan Zhou <zhoudongzhan@pjlab.org.cn>.

*Proceedings of the $42^{nd}$ International Conference on Machine Learning*, Vancouver, Canada. PMLR 267, 2025. Copyright 2025 by the author(s).

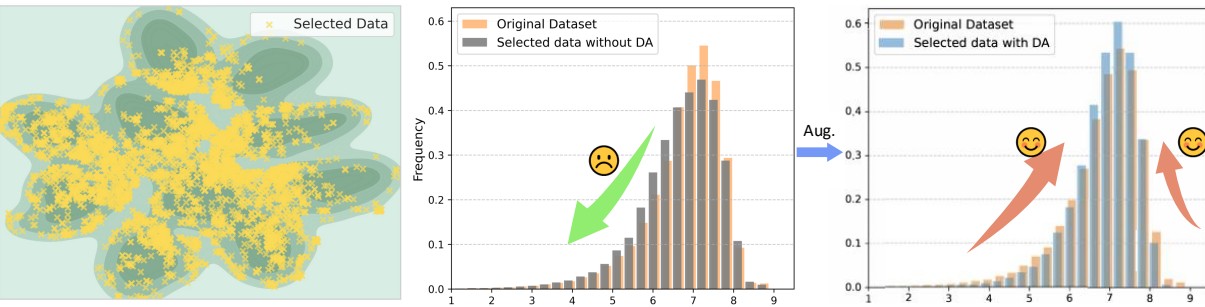

*Figure 1.* Illustration of the distribution of our selected data points using the T-SNE algorithm (*left*) and the density histograms without (*mid*) and with (*right*) augmenting the selected data on CIFAR-10. The selection ratio is 10%.

model learning and improves robustness. However, noisy or outlier samples typically exhibit relatively low density, increasing the risk of introducing noise. To address this, we introduce a semantic consistency distribution derived from the pre-trained multimodal model CLIP (Radford et al., 2021). By prioritizing samples with high sparsity and strong semantic alignment, our approach leverages the joint distribution of density and semantic consistency for effective and robust sample selection.

As illustrated in the left sub-figure of Fig. 1, the selected data points predominantly cluster around boundary regions among clusters. Meanwhile, the density histogram in Fig. 1 reveals a more balanced distribution after augmentation, with fewer low- and high-density samples and more data points converging toward the moderate-density regions. Compared to the distribution without DA, this redistribution highlights our framework's ability to enhance sparse regions, improving model generalization across the entire data distribution.

Experimental results across benchmark datasets and deep architectures demonstrate the effectiveness of our method. On large-scale datasets such as Tiny-ImageNet (Chrabaszcz et al., 2017) and ImageNet-1k (Deng et al., 2009), our method significantly accelerates training while maintaining or even improving generalization. For instance, on ImageNet-1k, our approach doubles the training efficiency while achieving comparable performance with the entire dataset. Moreover, our framework exhibits strong cross-architecture and cross-scenario generalization, effectively mitigating the impact of noisy data and enhancing versatility in real-world applications. On Tiny-ImageNet, our approach outperforms leading baselines by at least 3% in accuracy under noisy conditions, further demonstrating its reliability.

Our main contributions are summarized as follows: 1) We propose a novel training framework that dynamically integrates data selection and augmentation, significantly accelerating training while maintaining model performance. 2) We introduce a joint distribution based on density and semantic

consistency, ensuring effective sample selection and reducing noise and ambiguity. 3) Extensive experiments across diverse datasets and architectures demonstrate superior accuracy and generalization ability, particularly in noisy and challenging scenarios, validating its practical applicability.

## 2. Related Work

### 2.1. Data Selection

The primary goal of data selection is to enhance data-efficient learning, which can be broadly categorized into dataset distillation (Lei & Tao, 2023; Du et al., 2023; Sun et al., 2024; Liu & Wang, 2024), and static or dynamic data selection (or pruning) (Tan et al., 2024; Xia et al., 2023b; Sorscher et al., 2022; Qin et al., 2024). Dataset distillation focuses on synthesizing a small representative dataset that preserves the performance of training on the full dataset. In contrast, following dynamic data selection without synthesizing new data in this work, we propose a new data training framework that unifies dynamic data selection and augmentation to achieve enhanced model training acceleration.

**Static data selection** identifies a fixed subset of the training dataset before training begins. Existing methods can be categorized into selection with importance criteria, dataset distribution-based methods, and optimization-based methods. *Selection with importance criteria* computes per-sample importance scores and selects the most informative samples. This includes: 1) the expectation of $\ell_2$-norm error vector and the gradient norm (EL2N and GraNd) (Paul et al., 2021), 2) the change in the optimal empirical risk when a sample is removed (Tan et al., 2024), 3) the number of forgetting events in the whole training process (Forgetting) (Toneva et al., 2018), 4) the impact of including or excluding a sample on the model's classification ability (Feldman & Zhang, 2020). *Dataset distribution-based methods* select samples based on the geometric distribution of the dataset. Herding (Welling, 2009) chooses samples based on their distance from the corresponding class centers. D2 (Maharana et al., 2023) defines sample diffi-

culty by incorporating the difficulty of its neighboring examples. The work (Ramalingam et al., 2023) applies greedy k-center to select the coreset with good data coverage, and CCS (Zheng et al., 2023) balances the sample distribution and importance in selection. Similarly, Moderate-DS (Xia et al., 2023b) selects samples that are closer to the median score, aiming to balance diversity and representativeness. *Optimization-based methods* formulate selection as an optimization problem using techniques such as scalable self-supervised pruning metrics (Sorscher et al., 2022), influence function (Yang et al., 2023a), bi-level optimization (Killamsetty et al., 2021), gradient matching (Mirzasoleiman et al., 2020b), convex optimization (Mirzasoleiman et al., 2020a), facility location function (Yang et al., 2023b), temporal dual-depth scoring (Zhang et al., 2024), and submodularity (Iyer et al., 2021; Nohyun et al., 2023).

**Dynamic data selection** identifies informative samples throughout training, allowing the dataset to adapt as the model learns. The work (Raju et al., 2021) proposes UCB and $\epsilon$-greedy algorithms to estimate the uncertainty value associated with each training sample, selecting a subset of the data that exhibits the highest levels of uncertainty. Similarly, the work (He et al., 2024) also employs both prediction uncertainty and training dynamics to guide the selection process, ensuring that the most informative samples are retained throughout training. The work (Liu & Mirzasoleiman, 2022) proposes a data-efficient framework for training neural networks and achieves promising results. SAS (Joshi & Mirzasoleiman, 2023) improves data efficiency in SSL by proving and selecting the most beneficial data for contrastive training. Moreover, InfoBatch (Qin et al., 2024) proposes a method for unbiased dynamic data selection that accelerates training by pruning less informative samples to retain their relevance in model optimization, which allows for more efficient training without compromising the model's performance.

### 2.2. Data Augmentation

Data augmentation (DA) improves the generalization of deep neural networks by increasing the diversity of training samples (Yang et al., 2024a). Existing DA methods can be divided into image erasing/mixing-based and automatic augmentation methods (Xu et al., 2023). Image erasing and mixing-based augmentation erase some sub-regions in images or mix random information from two or more images for augmentation to create new samples, respectively. These methods include Cutout (DeVries & Taylor, 2017), RandomErasing (Zhong et al., 2020), HaS (Singh & Lee, 2017), AdvMask (Yang et al., 2022a), Mixup (Zhang et al., 2018), GuidedMixup (Kang & Kim, 2023), and GradSalMix (Hong et al., 2023), etc. In addition, based on pre-defined or optimized image transformation policies, automatic DA methods randomly apply one or multiple transformations to each

image at each epoch, including AutoAugment (Cubuk et al., 2019), Fast-AutoAugment (Lim et al., 2019), RandAugment (Cubuk et al., 2020), TrivialAugment (Müller & Hutter, 2021), SelectAugment (Lin et al., 2023), EntAugment (Yang et al., 2024b), and MADAug (Hou et al., 2023), etc. Beyond these, generative data augmentation further enriches data by synthesizing new samples using generative models (Moreno-Barea et al., 2020). Recent studies also emphasize representation consistency (Atienza, 2022) and address distribution gaps between clean and augmented data (He et al., 2019), pointing to new challenges in effective DA design.

## 3. The Proposed Method

### 3.1. Overview of the Proposed Method

As shown in Fig. 2, we propose a novel data training framework that integrates dynamic data selection with augmentation to enhance both training efficiency and generalization. Our framework employs two complementary distributions: 1). a density distribution, dynamically estimated by a task-specific model (e.g., ResNet), which identifies underrepresented samples for augmentation and training, and (2) a semantic consistency distribution, computed using a frozen pre-trained multimodal model (CLIP), which quantifies the alignment between an image and its corresponding textual label. Low-density regions highlight underrepresented samples but may also include noisy or ambiguous instances. To address this issue, the semantic consistency distribution acts as a strong complement, filtering samples with weak semantic alignment. By combining both distributions, we construct a joint distribution that captures the relationship between sample informativeness and semantic correctness. Consequently, samples with higher joint distribution scores are prioritized to be selected and augmented for training.

### 3.2. How to Identify Samples for Augmentation

Given a dataset $D$ that follows an underlying distribution $P(D)$, the optimization objective of our dynamic data selection module at time $t$ is to select a subset $\hat{D}_t$, containing at most $k$ samples. The goal is to minimize the expected loss over the distribution $P(D)$, with the following optimization: $\hat{D}_t = \underset{\hat{D}_t \subseteq D, |\hat{D}_t| \leq k}{\arg\min} \ \mathbb{E}_{z \sim P(D)} \left[ L\left(z, \hat{\theta}_{\mathcal{A}(\hat{D}_t)}\right) \right]$, where $z$ represents a test sample, $L$ is the loss function, $\hat{\theta}_{\hat{D}_t}$ is the empirical risk minimizer on $\hat{D}_t$, and $\mathcal{A}$ represents the augmentation operations applied to the selected subset.

Our method prioritizes low-density samples because such regions in the feature space often correspond to underlearned or insufficiently represented data points. Focusing on these samples and applying augmentation operations helps the model capture distinctive features. By augmenting these sparse samples, we compensate for their underrepresenta-

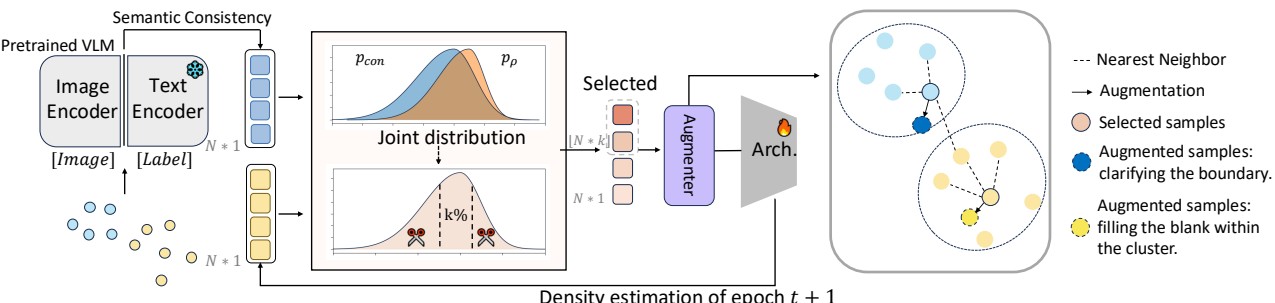

*Figure 2.* The framework of our proposed data training method: The core idea of our framework is to construct a joint distribution that integrates both the density and semantic consistency distributions, enabling the prioritization of low-density, semantically consistent samples. After augmentation, augmented sparse samples in intra-cluster regions help to fill the underrepresented spaces, while samples located around the decision boundaries between clusters differentiate the classification decision more clearly, thus improving generalization.

tion, improving the model's ability to generalize across diverse data regions. To efficiently determine the density of data points during online training, we exploit an online approximate nearest neighbor search architecture (HNSW) (Malkov & Yashunin, 2018) to query the nearest $k$ neighbors of each sample $x$, denoted as $NN(x)$. The density of a sample is then estimated as the mean of the $\ell_2$ distance between $x$ and its neighbors:

$$\rho_{x_i} = \frac{1}{k} \sum_{j \in NN(x_i)} ||x_i - x_j||, \tag{1}$$

where higher values indicate low-density samples. The density scores are then normalized using a Min-Max scaling to obtain $p_\rho(x)$. For augmentation, we employ slight augmentation, which generates neighboring samples within low-density areas while preserving local structure.

Nevertheless, low-density regions may also contain challenging, outlier, or noisy samples. Continuously selecting these samples for training can substantially complicate training, especially on real-world datasets that inevitably include noise. To address this issue and improve the practical effectiveness of our method, we introduce a multimodal semantic consistency constraint that simultaneously refines the selection of low-density samples. This ensures that augmentation is applied to meaningful data, improving both robustness and efficiency in the training process.

### 3.3. Multimodal Consistency Estimation for Robust Data Selection

Noisy data - arising from incorrect labels, corrupted images, or outlier samples - reflects a fundamental mismatch between the semantic content of $x$ and its corresponding label $y$. To detect and filter such inconsistencies, the data cleaner evaluates the joint distribution $p(x, y)$, which captures the plausibility of an image-label pair. Given the inherent correlation and multimodal nature of $x$ and $y$, we introduce

multimodal consistency supervision as an additional criterion for assessing sample reliability. This complements the density-based selector by filtering out samples that exhibit low cross-modal consistency.

To implement this, we leverage a pre-trained CLIP model to embed images and text into a shared multimodal space, enabling semantic alignment assessment. However, CLIP's zero-shot generalization is limited to domain-specific datasets, making it necessary to adapt the embeddings to the target domain. Instead of performing computationally expensive fine-tuning, we incorporate lightweight adapters of MLP for both the image and text encoders (Poth et al., 2023; Yang et al., 2024c). The lightweight architecture of adapters ensures efficient adaptation while preserving CLIP's pretrained knowledge.

To measure the cross-modal consistency, we compute the cosine similarity of the encoded image and text features:

$$con(x_i) = \ell_{cos}(E_I(x_i), E_T(y_i)), \tag{2}$$

where $E_I$ and $E_T$ are visual and textual encoders, respectively. The consistency scores are normalized via Min-Max scaling to approximate the consistency distribution $p_{con}(x)$, where higher values indicate stronger semantic alignment. Since the image-label alignment is derived from a pretrained vision-language model and remains independent of the training process, we precompute the consistency distribution beforehand. This enables direct use during sample selection, eliminating additional computational overhead in online training.

### 3.4. Augmenter

To integrate structural sparsity and semantic consistency, we define a joint distribution that combines both density and consistency distributions:

$$p_{sel}(x_i) = p_\rho(x_i) * p_{con}(x_i). \tag{3}$$

*Table 1.* The accuracy (%) comparison to state-of-the-art baselines. All methods are trained with ResNet-18 on CIFAR-10/100 and ResNet-50 on Tiny-ImageNet. Note that some results could not be computed due to the unavailability of open-source code and parameter settings, making it impossible to reproduce. Random* means randomly selecting samples in each epoch.

| Dataset | CIFAR-10 | | | CIFAR-100 | | | Tiny-ImageNet | | |
|---|---|---|---|---|---|---|---|---|---|
| Whole Dataset | 95.6 | | | 78.2 | | | 45.0 | | |
| Selection Ratio (%) | 30 | 50 | 70 | 30 | 50 | 70 | 30 | 50 | 70 |
| Random | 90.2 | 92.3 | 93.9 | 69.7 | 72.1 | 73.8 | 29.8 | 37.2 | 42.2 |
| Herding (Welling, 2009) | 80.1 | 88.0 | 92.2 | 69.6 | 71.8 | 73.1 | 29.4 | 31.6 | 39.8 |
| EL2N (Paul et al., 2021) | 91.6 | 95.0 | 95.2 | 69.5 | 72.1 | 77.2 | 26.6 | 37.1 | 44.0 |
| GraNd (Paul et al., 2021) | 91.2 | 94.6 | 95.3 | 68.8 | 71.4 | 74.6 | 29.7 | 36.3 | 43.2 |
| Glister (Killamsetty et al., 2021) | 90.9 | 94.0 | 95.2 | 70.4 | 73.2 | 76.6 | 30.1 | 39.5 | 43.9 |
| Forgetting (Toneva et al., 2018) | 91.7 | 94.1 | 94.7 | 69.9 | 73.1 | 75.3 | 28.7 | 33.0 | 41.2 |
| Moderate-DS (Xia et al., 2023b) | 91.5 | 94.1 | 95.2 | 70.2 | 73.4 | 77.3 | 30.6 | 38.2 | 42.8 |
| Self-sup. prototypes (Sorscher et al., 2022) | 91.0 | 94.0 | 95.2 | 70.0 | 71.7 | 76.8 | 27.7 | 37.9 | 43.4 |
| MoSo (Tan et al., 2024) | 91.1 | 94.2 | 95.3 | 70.9 | 73.6 | 77.5 | 31.2 | 38.5 | 43.4 |
| DP (Yang et al., 2023a) | 90.8 | 93.8 | 94.9 | - | 73.1 | 77.2 | - | - | - |
| Random* | 93.0 | 94.5 | 94.8 | 74.4 | 75.3 | 77.3 | 41.5 | 42.8 | 43.1 |
| UCB (Raju et al., 2021) | 93.9 | 94.7 | 95.3 | - | 75.3 | 77.3 | - | - | - |
| $\epsilon$-Greedy (Raju et al., 2021) | 94.1 | 94.9 | 95.2 | - | 74.8 | 76.4 | - | - | - |
| InfoBatch (Qin et al., 2024) | 94.7 | 95.1 | 95.6 | 76.5 | 78.1 | 78.2 | 42.2 | 43.2 | 43.8 |
| Ours | **94.9** | **95.5** | **96.0** | **77.6** | **78.9** | **79.5** | **44.9** | **47.0** | **49.4** |

Here, $p_\rho$ evolves dynamically with model training, allowing sample selection to adapt to the current model training state. During online training, samples with higher joint distribution scores are prioritized for augmentation and training, ensuring that both underrepresented and semantically meaningful samples are utilized effectively.

We employ TrivialAugment as our augmenter, which is widely used and offers a computationally efficient augmentation strategy. During augmentation, only a single lightweight transformation per image is applied. This brings two key advantages: i). It introduces negligible computation overhead to the online training process, making it highly efficient. ii). Since each image undergoes only one transformation with a slight magnitude, the augmented samples remain within their original local feature space. This is well-suited for the objective of our dynamic data selection framework: filling intra-cluster gaps and enhancing decision boundaries within clusters. Thus, the consistency of selected samples is preserved while the data diversity in sparse regions is enhanced. Consequently, training using these augmented samples improves model performance. In addition, we provide the augmentation operation list used in Table 9 in the Appendix, which includes image transformations such as translation, rotation, equalization, etc.

**Complexity analysis.** The computational costs of our framework, when integrated into online training, are primarily associated with density estimation. Specifically, both querying and updating within the HNSW graph operates with a complexity of $\mathcal{O}(\log(n))$, where $n$ is the total number of data points. Let $T$ denote the total number of training epochs; then, the total cost is $\mathcal{O}(T * \log(n))$. Since $T \ll n$, the overall computational complexity remains $\mathcal{O}(\log(n))$,

making our method scalable for large datasets. Furthermore, the data augmentation, as a standard pipeline in model training, introduces negligible overhead.

## 4. Experiment

### 4.1. Experiment Setup

**Datasets and network architectures.** In line with previous works (Tan et al., 2024; Xia et al., 2023b; Qin et al., 2024), we evaluate the effectiveness of our proposed method using widely adopted benchmark datasets, including CIFAR-10/100 (Krizhevsky et al., 2009), Tiny-ImageNet (Chrabaszcz et al., 2017), and ImageNet-1k (Deng et al., 2009). In addition, we evaluate the robustness of our method in noisy datasets. To further assess the generalization ability of our method, we extend the evaluation to more challenging datasets, such as ImageNet-A/O (Hendrycks et al., 2021b), ImageNet-Hard (Taesiri et al., 2024), and ImageNet-R (Hendrycks et al., 2021a). Additionally, we evaluate the generalization of our method across different deep architectures. Specifically, we conduct experiments using both ResNet-based, such as ResNet-18/50, and ViT-based models, such as ViT-B/L and Swin-Transformer, to demonstrate the robustness and scalability of our approach across diverse models.

**Comparison with state-of-the-arts.** We compare with our method both static and dynamic data selection methods, including 1) Random, 2) EL2N (Paul et al., 2021), 3) GraNd (Paul et al., 2021), 4) Herding (Welling, 2009), 5) Forgetting (Toneva et al., 2018), 6) Moderate-DS (Xia et al., 2023b), 7) Self-sup. prototypes (Sorscher et al., 2022), 8) MoSo (Tan et al., 2024), 9) DP (Yang et al., 2023a), 10)

*Table 2.* Results on ImageNet-1k with a 60% selection ratio using ResNet-50 on an 8-A100 server. Note that due to the high computational costs and device memory costs (Xia et al., 2023b), Glister and CG-Score are not reported. Some results are from (Qin et al., 2024). Time is the wall clock time; Overall (n*h) is the total GPU hour, where $n$ is the node number.

| Method | Herding | EL2N | GraNd | Forgetting | SSP | Moderate | MoSo | UCB | Infobatch | Glister | CG-Score | Ours | Whole Dataset |
|---|---|---|---|---|---|---|---|---|---|---|---|---|---|
| Acc. (%) | 71.1 | 72.3 | 71.0 | 72.5 | 70.0 | 73.1 | 74.0 | 75.8 | 76.5 | - | - | **76.9** | 76.4 |
| Time (h) | 10.5 | 10.5 | 10.5 | 10.5 | 10.5 | 10.5 | 10.5 | 10.5 | 10.5 | 10.5 | 10.5 | 10.5 | 17.5 |
| Overhead (h) | >17.5 | >17.5 | >17.5 | >17.5 | >24.0 | >17.5 | >17.5 | 0.03 | 0.0028 | - | - | 0.53 | 0.0 |
| Overall (n*h) | >224.0 | >224.0 | >224.0 | >224.0 | >276.0 | >224.0 | >224.0 | **84.0** | **84.0** | - | - | 88.2 | 140.0 |

*Table 3.* Experimental results on Tiny-ImageNet with noisy and corrupted data using ResNet-50. The noisy ratio is 20%.

| Method / | Noisy | | Corrupted | |
|---|---|---|---|---|
| Selection Ratio (%) | 20 | 30 | 20 | 30 |
| Random | 17.8 | 23.9 | 20.0 | 25.9 |
| Herding | 19.0 | 24.2 | 35.0 | 30.6 |
| Moderate-DS | 19.6 | 25.0 | 23.3 | 29.1 |
| EL2N | 13.9 | 18.6 | 18.6 | 24.4 |
| GraNd | 18.3 | 23.7 | 20.0 | 26.7 |
| Forgetting | 13.2 | 21.8 | 18.5 | 25.5 |
| Self-sup. prototypes | 15.1 | 21.0 | 20.2 | 26.9 |
| CG-Score | 8.4 | 15.3 | 16.4 | 24.4 |
| Glister | 21.6 | 25.5 | 21.2 | 22.0 |
| MoSo | 7.4 | 11.3 | 23.1 | 28.8 |
| Random* | 33.8 | 36.5 | 35.1 | 36.9 |
| InfoBatch | 34.9 | 37.1 | 35.1 | 38.1 |
| Ours | **35.9** | **39.6** | **39.1** | **42.0** |

*Table 4.* Experimental results on Tiny-ImageNet with data augmentation. The selection ratios are 30%, 50%, and 70%.

| Whole Dataset | | 52.0 | |
|---|---|---|---|
| Selection Ratio | 30% | 50% | 70% |
| Random | 29.8 | 37.2 | 42.2 |
| Herding | 31.6 | 39.2 | 45.6 |
| EL2N | 32.0 | 40.1 | 45.9 |
| GraNd | 32.2 | 40.5 | 46.2 |
| Glister | 33.1 | 42.2 | 46.5 |
| Forgetting | 27.2 | 36.2 | 44.2 |
| Moderate-DS | 33.8 | 41.5 | 46.6 |
| Self-sup. proto. | 33.4 | 41.1 | 46.6 |
| MoSo | 32.6 | 41.5 | 45.9 |
| Random* | 42.1 | 43.9 | 45.2 |
| InfoBatch | 43.2 | 45.9 | 48.3 |
| Ours | **44.9** | **47.0** | **49.4** |

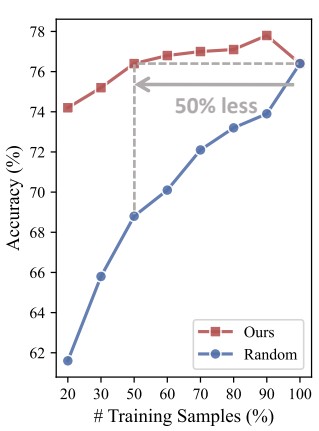

*Figure 3.* The performance on ImageNet-1k across various selection ratios with a 4-A100-GPU server.

UCB (Raju et al., 2021), 11) $\epsilon$-Greedy (Raju et al., 2021), 12) Glister (Killamsetty et al., 2021), and 13) InfoBatch (Qin et al., 2024).

**Implementation details.** To ensure consistency with prior work (Qin et al., 2024; Xia et al., 2023b), we follow similar experimental settings. Specifically, we use the OneCycle scheduler with the SGD/LARS optimizer for model training, a momentum of 0.9, a weight decay of 5e-4, and cosine annealing. We employ TrivialAugment (Müller & Hutter, 2021) in our framework. For fairness, we adopt the annealing and re-scaling techniques introduced in (Qin et al., 2024), which standardize the dynamic dataset pruning process across all methods compared. Moreover, we use InfoNCE loss to fine-tune adapters for 15 epochs on all datasets. Since InfoBatch uses soft pruning with a dynamic number of selected samples, we report its performance using the same number of forward passes as in our method.

### 4.2. Performance Comparison

As shown in Table 1, we evaluate the performance of our method by training ResNet-18 on CIFAR10/100 and ResNet-50 on Tiny-ImageNet across different selection ratios. Our method achieves comparable performance to models trained on the full dataset, even when only 50% of the data is used on CIFAR-10/100 and 30% on Tiny-ImageNet. In contrast, existing methods typically achieve lossless data selec-

tion with relatively higher selection ratios on these datasets, such as over 60% on CIFAR-10/100 and over 70% on Tiny-ImageNet.

Notably, our approach outperforms the other methods at the same selection ratios. On Tiny-ImageNet, a large-scale dataset, our method yields an average performance improvement of at least 2.7% while maintaining the same training costs. As the training data volume increases, this performance gap becomes even more pronounced, further highlighting the efficiency and effectiveness of our framework.

### 4.3. ImageNet-1k Results

Table 2 presents the evaluation results of our method on the ImageNet-1k dataset with a 60% selection ratio. Our approach outperforms the full dataset by achieving a nearly 40% training cost reduction, resulting in a reduction of up to 56 hours in training overhead with a 0.5% accuracy improvement. Meanwhile, since most static data selection methods require training surrogate models to determine the sample's influence throughout model training, the computation overheads are relatively much higher than ours. Thus, the results highlight that our method outperforms static data selection methods in both performance and computational efficiency. It also surpasses dynamic pruning methods in terms of final accuracy with comparable efficiency. These findings underscore the generality and competitiveness of

*Table 5.* Generalization of models trained with our method on ImageNet-Hard, ImageNet-A, ImageNet-R, and ImageNet-O. We report AUPR (%) for ImageNet-O and accuracy (%) on others. All models are ResNet-50.

| Selection Ratio(%) | 20 | 30 | 50 | 60 | 70 | 80 | 90 | Whole Dataset |
|---|---|---|---|---|---|---|---|---|
| ImageNet-A | 1.9↓1.2 | 2.1↓1.0 | 2.9↓0.2 | 3.1↑0.0 | 3.4↑0.3 | 3.4↑0.3 | **3.5↑0.4** | 3.1 |
| ImageNet-R | 37.2↑1.0 | 38.5↑2.3 | 39.3↑3.1 | 39.8↑3.6 | 39.9↑3.7 | 40.6↑4.4 | **41.0↑4.8** | 36.2 |
| ImageNet-O | 15.4↑2.2 | 15.8↑2.6 | 16.1↑2.3 | 16.3↑2.5 | 16.3↑2.5 | 16.4↑2.6 | **16.5↑2.7** | 13.2 |
| ImageNet-Hard | 14.2↓0.5 | 15.3↑0.6 | 15.9↑1.2 | 16.5↑1.8 | 16.7↑2.0 | 17.2↑2.5 | **17.5↑2.8** | 14.7 |

our approach to large-scale datasets.

Further analysis of the performance across different selection ratios on ImageNet-1k is shown in Fig. 3. The results show that our method achieves lossless performance with only 50% of the training data. When using 20% of the data, performance drops by about 2% while nearly 80% of the training overhead is eliminated. Compared to random selection, which suffers a significant accuracy drop as the selection ratios decrease, our method maintains robust performance even with reduced data. Similarly, most existing baseline methods typically require at least 60% of the training data to achieve similar lossless performance. This demonstrates that our framework further lowers the data requirement for maintaining full performance.

### 4.4. Robustness to Noisy Scenarios

In real-world scenarios, training data is often polluted by corrupted and mislabeled images (Xia et al., 2023a; Wang et al., 2018), which can significantly degrade model performance. Specifically, we simulate mislabeled data (noisy) by flipping a portion of the labels to incorrect ones using symmetric label noise. Meanwhile, we introduce five types of realistic distortions to simulate corrupted data, namely Gaussian noise, random occlusion, resolution variations, fog, and motion blur. The examples of corrupted data are shown in the Appendix. To evaluate the practical relevance of our data training framework in such noisy environments, we assess the robustness of our method compared to existing state-of-the-art methods.

As shown in Table 3, our approach consistently outperforms the compared methods, demonstrating superior robustness against both mislabeled and corrupted data. Specifically, our method achieves a 4% improvement over competing methods on corrupted datasets, even with a 20% noise ratio on Tiny-ImageNet. Our framework excels in these scenarios due to its ability to combine low-density sample selection with multimodal semantic alignment. While noisy data is typically sparse and low-density, our method's robust integration of multimodal semantics offers a powerful mechanism for mitigating noise and highlighting meaningful patterns in the data. This approach allows us to maintain high robustness without sacrificing data efficiency.

By prioritizing sparse, low-density samples and leveraging

the corrective power of multimodal alignment, our method provides a reliable and efficient solution for robust deep learning in practical, noisy data environments, demonstrating its practical significance.

### 4.5. Effect of Data Augmentation on Model Performance

To further assess the effectiveness of our method, we compare its performance against several baseline methods using TrivialAugment for data augmentation, as shown in Table 4. Our method consistently outperforms the other approaches across various selection ratios.

While data augmentation enhances the performance of all methods, our approach consistently achieves superior results at different selection ratios. This indicates that our method is not simply a straightforward combination of data augmentation and data selection. Instead, it effectively identifies the most beneficial samples for augmentation, leading to significant performance improvements. By selectively amplifying the impact of data augmentation, our method optimizes model performance, demonstrating its ability to leverage augmentation more effectively than other approaches.

### 4.6. Generalization on Hard Benchmarks

To evaluate the generalization capabilities of our proposed framework, we conduct experiments on challenging benchmark datasets, including ImageNet-Hard (Taesiri et al., 2024), ImageNet-R (Hendrycks et al., 2021a), and ImageNet-A/O (Hendrycks et al., 2021b). Specifically, we pre-train ResNet-50 models using data selected through our method across various selection ratios and then test their performance on these challenging benchmark datasets. Following standard evaluation settings, we report the area under the precision-recall curve (AUPR) for ImageNet-O and classification accuracy for the other datasets.

As shown in Table 5, our method maintains or even enhances generalization performance on these challenging datasets, despite using fewer training samples. The results demonstrate that reducing the dataset size with our framework does not compromise generalization ability. Meanwhile, as the selection ratios increase, our method achieves superior generalization compared to training on the full dataset.

*Table 6.* Experiment results on more advanced architectures, including ViT-B, ViT-L, and Swin-T on ImageNet-1k with a 4-A100 GPU server. Overhead represents GPU hours (h), and $S_r$ refers to the selection ratio.

| $S_r(\%)$ | 50 | 60 | 70 | 80 | 90 | Full Dataset |
|---|---|---|---|---|---|---|
| ViT-B | 82.6 | 82.9 | 83.2 | 83.2 | **83.3** | 82.5 |
| ViT-L | 85.2 | 85.3 | 85.6 | **85.7** | **85.7** | 84.6 |
| Swin-T | 84.1 | 84.1 | 84.2 | 84.2 | **84.3** | 84.2 |

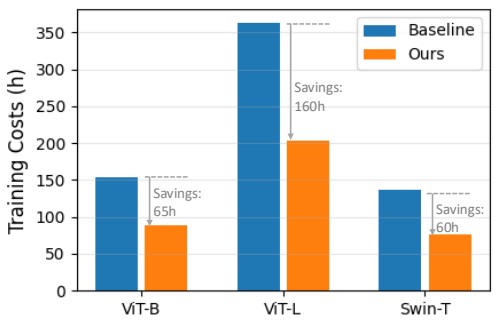

*Figure 4.* The overall cost savings achieved by our method on ViT-based architectures with lossless performance. The experiment is conducted on ImageNet-1k with a 4-A100-GPU server.

### 4.7. Generalization on Different Architectures

To evaluate the scalability of our proposed method, we conduct experiments on advanced architectures, including ViT-B, ViT-L (Dosovitskiy et al., 2020), and Swin-T (Liu et al., 2021). Specifically, we train these architectures using our framework across various selection ratios.

As shown in Table 6, our framework is architecture-agnostic, achieving robust generalization across these different models, even with reduced selection ratios. Notably, the lossless performance can be achieved with only 50% of the training data. The results underscore that our method generalizes on ResNet-based and Transformer-based architectures, all with reduced training costs.

Additionally, in Fig. 4, we present the practical training costs on these architectures and the lossless cost savings achieved by our framework. It can be seen that our proposed framework can significantly save hundreds of hours on large-scale architecture training.

### 4.8. Visualization of the Selection Robustness

In Fig. 1, we illustrate our selection results on clean datasets, showing that the selected data points mainly cluster around boundary regions among clusters. To better understand our selection effectiveness, in Fig. 5, we further illustrate our selection results on the noisy Tiny-ImageNet dataset with a 20% noise ratio. It can be seen that compared to the baseline, our method can effectively filter out noisy samples:

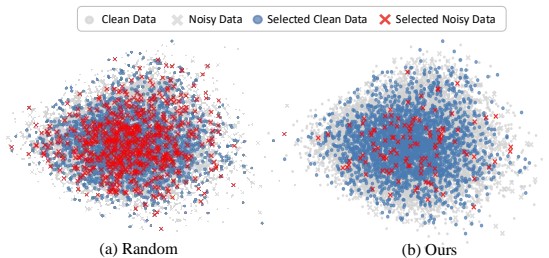

(a) Random (b) Ours

*Figure 5.* Visualization of the selection results on noisy Tiny-ImageNet with a 20% noise ratio. The selection ratio is 20%.

the number of selected noisy points is minimized.

### 4.9. Further Analysis of the Overheads

Although our method introduces negligible overheads into online training, our framework incorporates adapter fine-tuning and feature embedding via CLIP models and corresponding adapters before model training begins. As shown in Table 7, we analyze these pre-computation overheads of the adapter fine-tuning and feature embedding via CLIP models. It can be observed that these one-time overheads before model training are negligible compared to standard target model training. Once computed, no further computation is required during online training across selection ratios.

### 4.10. Ablation Study

**Effect of different modules in our framework.** In Table 8, we systemically evaluate the effectiveness of different components within our proposed framework on Tiny-ImageNet using ResNet-50 across various selection ratios.

When only the density distribution $p_\rho$ is used, performance is lower, as low-density samples often include sparse and outlier data, which can introduce ambiguity into the training process. However, when both the density distribution $p_\rho$ and consistency distribution $p_{con}$ are combined, performance improves, demonstrating that incorporating semantic consistency helps mitigate the negative effects of density-based selection. Further performance gains are achieved by including the augmentation module, which boosts accuracy by a significant margin. This shows that augmentation plays a crucial role in improving performance, especially when the selected data points under $p_\rho$ and $p_{con}$ are more suited for augmentation, enhancing the model's generalization. The results indicate that removing any module from our framework leads to a substantial drop in performance.

## 5. Conclusion

This paper proposes a novel data training framework that unifies dynamic data selection and data augmentation for

*Table 7.* Overheads of fine-tuning and feature embedding before model training on large-scale datasets with a 1-V100 GPU server.

| Dataset | Fine-tuning | Embedding | Overall training |
|---|---|---|---|
| Tiny-ImageNet | 0.39h | 0.03h | 21.0h |
| ImageNet-1k | 1.25h | 0.17h | 84.0h |

*Table 8.* Effect of density distribution, consistency distribution, and augmenter on Tiny-ImageNet using ResNet-50. We report test accuracy (%). The selection ratios (%) are 30%, 50%, and 70%.

| $p_\rho$ | $p_{con}$ | aug. | 30% | 50% | 70% |
|---|---|---|---|---|---|
| ✓ | | | 39.0 | 40.7 | 42.5 |
| | ✓ | | 42.0 | 45.6 | 45.8 |
| | | ✓ | 41.6 | 45.9 | 48.3 |
| | ✓ | ✓ | 42.5 | 46.3 | 49.3 |
| ✓ | ✓ | | 41.5 | 43.1 | 44.3 |
| ✓ | | ✓ | 41.1 | 45.1 | 48.5 |
| ✓ | ✓ | ✓ | **43.5** | **47.5** | **50.2** |

more enhanced model training acceleration. Unlike existing selection methods, our proposed approach identifies samples suitable for data augmentation. By combining this with augmentation, our framework can improve model generalization with reduced training costs. As a result, we can achieve lossless training acceleration with fewer data and enhanced generalization using the same volume of data. Extensive experiments demonstrate the effectiveness and efficiency of our method, especially in terms of generalization across large-scale datasets and more challenging scenarios.

## Acknowledgement

This work is supported by the STI 2030-Major Projects of China under Grant 2021ZD0201300, the Fundamental Research Funds for the Central Universities under Grant 2024300394, the National Natural Science Foundation of China under Grant 62276127. This work is supported by the Shanghai Municipal Science and Technology Major Project. This work is supported by Shanghai Artificial Intelligence Laboratory.

## Impact Statement

We propose a unified framework for dynamic data selection and augmentation that improves training efficiency and generalization in deep learning. By reducing data and computational requirements, our method promotes more sustainable and accessible AI development, especially for large-scale models and resource-constrained settings.

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

# A. More Implementation Details

## A.1. Augmentation Space

*Table 9.* Augmentation operations and their magnitudes.

| Operation | Value Range | Magnitude-based |
|---|:---:|:---:|
| Identity | - | ✗ |
| ShearX | [0.0, 0.99] | ✓ |
| ShearY | [0.0, 0.99] | ✓ |
| TranslateX | [0.0, 32.0] | ✓ |
| TranslateY | [0.0, 32.0] | ✓ |
| Rotate | [0.0, 135.0] | ✓ |
| Brightness | [0.0, 0.99] | ✓ |
| Color | [0.0, 0.99] | ✓ |
| Contrast | [0.0, 0.99] | ✓ |
| Sharpness | [0.0, 0.99] | ✓ |
| Posterize | [2, 8] | ✓ |
| Solarize | [255.0, 0.0] | ✓ |
| AutoContrast | - | ✗ |
| Equalize | - | ✗ |

# B. Discussion and Future Work

In this paper, we propose a novel online data training framework that unifies dynamic data selection and data augmentation to achieve enhanced model training acceleration. In this section, we discuss some potential limitations and future work for our method. 1). Our proposed method is based on the pretrained CLIP model to estimate the sample semantic consistency. While this exhibits superior effectiveness in general datasets, further applying our method to special tasks, such as medical imaging, where pretrained multimodal models are unavailable or mismatched, is worth exploring in future work. 2). We employ our framework on image classification tasks and demonstrate its superior effectiveness. Future work should extend the applications to more real-world tasks, such as object detection and segmentation.

