# OpenReview forum: "When Dynamic Data Selection Meets Data Augmentation: Achieving Enhanced Training Acceleration"
_ICML.cc/2025/Conference — ICML 2025 poster_

### Official Review · Reviewer_3S6N · 2025-03-03

**Overall Recommendation:** 4

**Summary:**

Paper centers around the idea of combining dynamic data selection and augmentation in order to increase both quality and diversity of data. The proposed method is applicable specifically to multimodal data. The proposed methods selects augmentation candidates that are both low density and do not represent noisy outliers. The later selection criteria is fulfilled by focusing on the examples for which the samples' and labels' representations align (cos similarity of embeddings) according to a CLIP model fine-tuned on the original (pre-selection&augmentation) data. Authors demonstrate the effectiveness and efficiency of the proposed method and its components in various comparisons to various baselines.

**Claims And Evidence:**

The central claim of the paper is: selecting samples according to the proposed methodology improves model generalization with reduced training cost. The claim seem to be justified overall. Yet, I still have some doubts and questions regarding the experiments presented in the paper, see following sections.

**Essential References Not Discussed:**

Given that the focus is image & multimodal image-text domains, the references are discussed sufficiently well to the best of my knowledge.

**Experimental Designs Or Analyses:**

I checked the experimental design and analysis. I have the following questions and I apologize if I have overlooked something and the answers are already in the manuscript:
- Is the only method that includes augmentation in Table 1 the methods presented in this paper? I would like to see at least the result of Random + augmentation (and optimally InfoBatch + augmentation) on all datasets (Table 4 only shows it for Tiny-ImageNet)
- In the presented experiments, did authors fine-tune a pre-trained CLIP model with the selection&augmentation methods appleid? Or was training from scratch used?
- iiuc authors have to fine-tune a CLIP model that is used for semantic alignment estimation on the data prior to filtering. It would be interesting to include the performance of this model in the results.
- is the cost of fine-tuning CLIP included in the results presented in Table 2?
- for Table 3, I would be interested in seeing the performance of top 2 methods with augmentation (i.e. in Table 3 also the only method using augmentation is the one presented in this work?)
- when talking about selection ratio, e.g. Figure 3 x-axis, Table 4 etc., the ratio does not include the augmented examples, doe it? i.e. the numb er of examples the model is trained on is actually larger than e.g. 20% of the full dataset for selection ratio = 20%? What is the augmentation ratio (i.e. how many new examples are generated per selected example in the presented experiments?)

**Methods And Evaluation Criteria:**

Yes, they make sense overall.

**Other Comments Or Suggestions:**

- typo in line 088 second column "riak" -> "risk"
-  can authors discuss how applcable this method would be to other domains like e.g. data selection for LLM training?

**Other Strengths And Weaknesses:**

I would like to see the experimental section as the main strength of this paper. The experiments are telling a nice story and are well designed in order to justify the main claims of the paper. However, given the various questions I had while reading this section (see experimental design section), it is also the main weakness of this paper at this point.

**Questions For Authors:**

See "experimental design" section.

**Relation To Broader Scientific Literature:**

While the title and the introductions are formulated rather generally, the paper's focus is very much on multimodal image-text data. Hence, the focus of the related works section is on multimodal or image only domains. I suggest authors make it more clear already in the intro that the focus of this work is on multimodal data specifically.

**Theoretical Claims:**

No theoretical claims.

---

> ### Author Rebuttal · Authors · 2025-03-31
>
> Dear reviewer 3S6N,
>
> Thank you for your careful review and constructive suggestions on our work.  We appreciate your recognition of our work's strengths, e.g, novelty, well-structured experiments.
>
> We provide responses to address the comments as follows:
>
> - **Q1: More results in Table 1.**
> - **A1:** Thank you for your suggestions. We have extended our evaluation to include more results in Tab. 1 (as Tab. 4 shows results on Tiny-ImageNet).
>
>     As shown in Tab. D-1, while both baselines benefit from augmentation, our method consistently achieves higher accuracy. This indicates that the performance gains are not solely due to augmentation but stem from our method's ability to identify samples suitable for augmentation. Thus, by unifying dynamic data selection and augmentation, our framework achieves both training acceleration and enhanced performance.
>
> **Table D-1:** Additional results for Tab. 1.
> ||C-10|||C-100|||
> |-|-|-|-|-|-|-|
> |Selection ratio (%)|30|50|70|30|50|70|
> |Random*|93.3| 94.8| 95.2|76.8|77.9| 78.5|
> |Infobatch|94.8|95.3| 95.9|77.1|78.5 |78.7|
> |Ours|**94.9**|**95.5**|**96.0**|**77.6**|**78.9**|**79.5**|
>
>
> - **Q2: Clarification on the fine-tuning process.**
> - **A2:** We would like to clarify that the fine-tuning process does **NOT** use the selection\&augmentation methods. The CLIP backbone is kept frozen; we only train the lightweight adapter from scratch.
>
> - **Q3: Performance of the fine-tuned model.**
> - **A3:** Good question. First, in Tab. D-2, we show the accuracy of CLIP before and after fine-tuning.
> It shows that the fine-tuned CLIP achieves notable improvements, indicating enhanced dataset-specific alignment ability.
> Second, we employ the pretrained CLIP without fine-tuning in our method. In Tab. D-3, while the vanilla CLIP achieves high performance, the fine-tuned model further improves the performance.
>
> **Table D-2:** Performance of CLIP in classification accuracy.
> ||C-10|C-100|
> |-|-|-|
> |zero-shot|89.8|65.1|
> |fine-tuned|**94.8**|**76.8**|
>
> **Table D-3:** Performance of CLIP with Tiny-ImageNet.
> |Selection ratio (%)|30|50|70|
> |-|-|-|-|
> |w/o fine-tuning| 40.2|45.7|48.5|
> |Ours| **44.9**|**47.0**|**49.4**|
>
> - **Q4: Clarification on the cost in Tab. 2.**
> - **A4:** Yes, the cost of fine-tuning is **included** in the total costs reported in Tab. 2 to ensure a fair comparison with other methods.
> We freeze the CLIP backbone and only fine-tune a lightweight adapter (a simple linear layer constituting only 0.04% of CLIP ViT-B/32’s parameters) with minimal training iterations. Thus, the fine-tuning process can be completed efficiently.
>
> - **Q5: More results for Tab.3.**
> - **A5:** Thank you for the suggestion.
> As shown in Tab. D-4, our method maintains superior performance across both settings and selection ratios. Notably, Infobatch performs worse than the random baseline in some cases in noisy conditions. This is because it prioritizes high-loss samples, which are often noisy or corrupted under noisy scenarios. Applying augmentation to noisy samples further exacerbates their negative impact as more misleading training signals are introduced, ultimately degrading performance.
>
>     This highlights that our performance gain is not solely due to augmentation, but rather due to selecting the semantic-representative samples for training.
>
> **Table D-4:** Additional results for Table 3.
> ||Noisy||Corrupted||
> |-|-|-|-|-|
> ||20|30|20|30|
> |Random*|34.8|37.9|36.0|39.1|
> |Infobatch|34.9|37.1|35.1|38.1|
> |Ours|**35.9**|**39.6**|**39.1**|**42.0**|
>
> - **Q6: Clarification on the selection ratio.**
> - **A6:** We would like to clarify that, following common practice in data augmentation and selection research, our method trains models **only on the augmented data**. In each epoch, we generate **exactly one augmented sample per selected data point**, and only these augmented samples are used in both forward and backward passes. Thus, the selection ratio is exactly equal to the proportion of the data used for training.
> - **Q7: Clarification of multimodal focus in the Introduction.**
> - **A7:** Thank you for the suggestion. We will revise the intro part, clarifying the multimodal characteristic and focus of our framework.
> - **Q8: Typo.**
> - **A8:** We have corrected the mentioned typo and thoroughly reviewed the manuscript to ensure clarity and correctness.
> - **Q9: Applicability to LLM training.**
> - **A9:** Applying our method to other domains, such as data selection for LLM training, is a promising direction for future research. Since our current work focuses on the image and image-text domains, extending this framework to LLM training requires adapting the definitions and estimation for both semantic consistency and sample density in a purely textual feature space. Feasible solutions may involve leveraging pre-trained language models (e.g., BERT, GPT) to assess semantic alignment and correctness, and using task-specific metrics such as token-level uncertainty or embedding-space sparsity to approximate density distributions.

---

### Official Review · Reviewer_GA1s · 2025-03-10

**Overall Recommendation:** 4

**Summary:**

The authors propose a unified framework combining dynamic data selection and data augmentation to accelerate model training. An online nearest neighbour search is used to find low-density samples along with a semantic consistency score from a pre-trained CLIP model to filter out noisy data. The targeted augmentation of the filtered data helps fill data gaps and sharpen decision boundaries in sparse regions, improving model generalization and robustness. The authors claim to achieve lossless training acceleration with fewer data points using a similar amount of data.

## update after rebuttal
The authors have sufficiently addressed all my concerns and included more analyses to show the strength of their method. I have accordingly increased my score as a result.

**Claims And Evidence:**

Most of the claims made by the authors are supported by clear and convincing evidence. From the results, integrating dynamic data selection with targeted augmentation seems to improve efficiency and generalization. One claim that needs a bit more validation though is the universal applicability of the method to all domains or tasks. The reliance on a pre-trained CLIP model for semantic consistency might not transfer well to domains where such models are less effective. It may also be useful to study the benefits and limits of the method under extreme noise conditions (both label and input noise) which might affect the augmenter part of the framework.

**Essential References Not Discussed:**

It may be useful to mention generative methods of data augmentation:

Moreno-Barea, F.J., Jerez, J.M. and Franco, L., 2020. Improving classification accuracy using data augmentation on small data sets. Expert Systems with Applications, 161, p.113696.

Some more recent works that talk about improving generalization by agreement and the gap between clean and augmented data:

Atienza, R., 2022. Improving model generalization by agreement of learned representations from data augmentation. In Proceedings of the IEEE/CVF winter conference on applications of computer vision (pp. 372-381).

He, Z., Xie, L., Chen, X., Zhang, Y., Wang, Y. and Tian, Q., 2019. Data augmentation revisited: Rethinking the distribution gap between clean and augmented data. arXiv preprint arXiv:1909.09148.

**Experimental Designs Or Analyses:**

The experimental design and analyses seem sound and valid. The authors include extensive experiments to show evidence that their method outperforms several baselines. They include results showing that models trained across different selection ratios achieve comparable performance to models trained on the full dataset and the proposed method has one of the lowest computational costs. They also report results showing that their method works on several datasets and models, including more advanced architectures. They show the method's robustness to noise and perform an ablation study on the density distribution, consistency distribution, and augmenter to show that all three are necessary for the best results.

The authors mention that they leverage a pre-trained CLIP model to embed images and text into a shared multimodal space for semantic alignment assessment. They mention that they use lightweight adapters to adapt embeddings to target domains, however, if there are target domains where CLIP-like models or adapters are less effective, this method may not work too well since this is a major part of the framework.

**Methods And Evaluation Criteria:**

The proposed methods and evaluation criteria make sense for the problem at hand. The authors use the CIFAR-10, CIFAR-100, Tiny-ImageNet, and also the ImageNet-Hard, ImageNet-A, ImageNet-R, and ImageNet-O datasets. They use the ResNet-18 and ResNet-50 models along with more advanced architectures such as ViT-B, ViT-L, and Swin-T to show the strength of their method. The selection ratio and accuracy is reported across all of these experiments.

**Other Comments Or Suggestions:**

Minor typos:
- 012: "lossless performances" -> "lossless performance"
- 042: "reinforces the model learning" -> "reinforces model learning"
- 088: "riak" -> "risk"
- 101: unclear "balances the distribution and the sample distribution and importance in selection"
- 104: "Optimization-based methods formulates" -> "Optimization-based methods formulate"
- 645: "occlution" -> "occlusion"

**Other Strengths And Weaknesses:**

Strengths:
- Novel combination of the existing ideas of data selection and data augmentation to come up with data selection for data augmentation.
- Extensive experiments across various models, datasets, and selection ratios.
- Robustness to noise.
- Low computational costs.

Weaknesses:
- The use of CLIP-like models in the framework which might not transfer to all tasks and domains.
- Might be good to study the method's limits under extreme noise conditions (label and input noise).
- A few typos (mentioned below).

**Questions For Authors:**

1. How exactly is $p_{sel}$ understood as the joint distribution that combines both density and consistency distributions? The definition and explanation need more clarity - it is defined but never mentioned again.
2. Have you tested the sensitivity of your approach to the choice of the pre-trained model or explored its applicability in domains where CLIP or the available pre-trained models/adapters might not perform well? More details on the pre-trained models and adapters would be very beneficial.

Addressing these questions sufficiently would improve the strength of this method significantly.

**Relation To Broader Scientific Literature:**

The key contributions of this work are in combining the existing ideas of data selection and data augmentation to pick the best data to augment. Previous data selection methods are more static and might not find low-density samples. Previous data augmentation methods are not as targeted and might not fill in sparse gaps in the training distribution. Combining these two areas to both identify underrepresented samples while selecting ones that are best for augmentation fills a gap in the literature while reducing training costs and enhancing robustness to noise.

**Theoretical Claims:**

There are no major theoretical claims or proofs in this paper.

---

> ### Author Rebuttal · Authors · 2025-03-31
>
> Dear reviewer GA1s,
>
> Thank you for your meticulous review and valuable suggestions on our work. We appreciate your recognition of our work's strengths, e.g., novelty, sound and valid experiments.
>
> We provide responses to address the comments as follows:
>
> - **Q1: Universal applicability and reliance on CLIP.**
> - **A1:** Thank you for pointing this out. We agree that the applicability and effectiveness of pretrained CLIP may vary across domains, particularly in some highly specialized fields.
> However, we note that this concern is not unique to our work but is shared by many state-of-the-art works that leverage pretrained VLM such as CLIP.
> Since our work focuses on dynamic data selection in general-purpose vision domains, fully addressing CLIP’s applicability across all domains and tasks is, in our opinion, beyond the scope of this work but of interest for our future work.
> Some feasible solutions may include leveraging fine-tuned VLM tailored to the target domain, or incorporating domain-specific adapters or alignment modules to improve semantic consistency estimation in more specialized scenarios.
>
>     We will add this discussion to the main paper on page 8 before the Conclusion section.
>
> - **Q2: Extreme noise conditions.**
> - **A2:** Thank you for the suggestion. To assess the robustness of our method under extreme noise conditions, we conducted experiments on Tiny-ImageNet with high label, input noise, and both (with 50% and 70% noise ratio).
>
>     As shown in Table C-1, while the overall performance naturally degrades under severe noise, our approach consistently achieves significantly higher accuracy, e.g., over 10% accuracy improvement in the presence of input noise. Moreover, we further analyze the average proportion of noisy samples retained during training (Table C-2). It shows that the proportion of noisy samples in our selected datasets remains considerably low even in high-noise conditions. These results underscore the benefits of our approach: even in highly noisy environments, the multimodal semantic consistency used for sample selection helps retain informative, high-quality data and maintain model robustness.
>
> **Table C-1**: Comparison of accuracy with Tiny-ImageNet in high-noise conditions.
> |Noise Ratio (%)|50||70||
> |-|-|-|-|-|
> |Selection Ratio (%)|20|30|20|30|
> |**Label Noise**||||
> |Random*|25.5|26.1|17.6|18.0|
> |Ours|**27.6**|**30.7**|**19.2**|**21.8**|
> |**Input Noise**||||
> |Random*|28.5|28.1|18.6|19.0|
> |Ours|**38.3**|**40.9**|**35.6**|**39.4**|
> |**Label\&Input Noise**||||
> |Random*|24.1|25.2|14.2|15.9|
> |Ours|**26.1**|**28.1**|**16.2**|**18.5**|
>
> **Table C-2**: Further analysis of the label noise proportion (%). We report the average introduced noise ratio in the selected datasets through the entire training process.
> |Noise Ratio (%)|50||70||
> |-|-|-|-|-|
> |Selection Ratio (%)|20|30|20|30|
> |Random|20.3|30.5|20.1|29.9|
> |Ours|**3.1**|**5.7**|**4.0**|**6.1**|
>
>
> - **Q3: More suggested related works.**
> - **A3:** We will include the suggested works in Sec. 2.2 of the revised version. Specifically,
>
>      - Sec 2.2 line 152: add references "*Beyond these, generative data augmentation (Moreno-Barea et al., 2020) ...*" and "*Recent studies also emphasize representation consistency (Atienza, 2022) and address distribution gaps between clean and augmented data (He et al., 2019) ...*".
>
>
> - **Q4: Minor Typos.**
> - **A4:** We have corrected all the mentioned issues and thoroughly reviewed the entire manuscript to ensure clarity and correctness in the final version.
>
>
> - **Q5: Explanation of $p_{sel}$.**
> - **A5:** As defined in Eq. (3) of Sec. 3.4, $p_{sel}$ is computed as the product of $p_\rho$, which reflects the density distribution, and $p_{con}$, which captures semantic consistency.
> Specifically, $p_\rho$ assigns higher values to low-density samples, promoting coverage of underrepresented regions in the feature space.
> $p_{con}$ prioritizes samples with strong semantic alignment, ensuring their informativeness and relevance
> By combining the two, $p_{sel}$ encourages samples that are both structurally important and semantically meaningful.
>
>     We appreciate your suggestion and will revise Sec. 3.4 to include this clarification.
>
>
> - **Q6: Sensitivity to the choice of pre-trained model.**
> - **A6:** Insightful question. To assess the sensitivity of our method to the choice of pre-trained models, we conduct experiments replacing CLIP with another pretrained multimodal model LanguageBind (LB)[a], which focuses on aligning diverse modalities (e.g., video, audio, infrared, and depth) into a shared language space.
> While CLIP is highly optimized for image-text alignment estimation, our framework still achieves consistent performance when using LB, with only a marginal drop (< 0.7%) in Table C-3.
>
> **Table C-3**: Sensitivity analysis on the pretrained model.
> |Selection Ratio (%)|30|50|70|
> |-|-|-|-|
> |Random*|41.5|42.8|43.1|
> |CLIP|44.9|47.0|49.4|
> |LB|44.2|46.5|48.7|
>
> [a] Zhu, Bin, et al. Languagebind, ICLR'24.

---

> > ### Comment · Reviewer_GA1s · 2025-04-01
> >
> > Thanks to the authors for the thorough response to all of my points! I will increase my recommendation accordingly.

---

> > > ### Author Response · Authors · 2025-04-02
> > >
> > > Dear reviewer GA1s,
> > >
> > > We would like to express our sincere gratitude to reviewer GA1s for acknowledging our work and providing constructive suggestions. Thanks again for the time and effort in reviewing our work.

---

### Official Review · Reviewer_bQXA · 2025-03-14

**Overall Recommendation:** 2

**Summary:**

Data selection---eliminating unhelpful samples---plays a crucial role in machine learning. While selecting high-value samples improves training efficiency without degrading performance, it can reduce data diversity and harm model generalization. To address this, this paper proposes a unified framework that integrates data selection and data augmentation, achieving both reduced computational cost and enhanced performance. Moreover, instead of naively combining these two techniques, they estimate each sample's local density and semantic alignment in a joint distribution with the help of multimodal model CLIP. Empirical results show that our approach accelerates training while improving model generalization.

**Claims And Evidence:**

1. I agree that data selection can improve training efficiency and maintain comparable performance, but it inherently **reduces data diversity, which could impact generalization**. However, the claim that it harms model generalization lacks supporting evidence. Are there any preliminary results to validate this statement? And how to set up such experiments to present or measure model generalization.

2. In the proposed framework, if a data point is selected for augmentation, is the model trained on both the original and augmented versions, or only on the augmented one? From my understanding, only the augmented version is used, but wouldn’t retaining the original sample help preserve semantic completeness?

3. The proposed method does not explicitly control the outcomes of data augmentation. How can we ensure that post-augmentation samples maintain strong semantic alignment with the original data points and do not negatively impact model training? Introducing a verification mechanism for post-processing might be beneficial.

4. Furthermore, if the augmentation process preserves local structure (as mentioned in lines 210–211, right-hand side), is it necessary to continuously update the feature space during training to identify low-density samples? How does the relationship between a sample and its neighbors change before and after augmentation?

**Essential References Not Discussed:**

There are no major additional references to include, as the baseline methods used for comparison are commonly employed in the data selection literature.

**Experimental Designs Or Analyses:**

Yes, their experimental design makes sense to me. However, in their setup, the CLIP model and a ResNet-18 model are used as **embedding generators for filtering**, while the model being trained is from the **ViT series**. Could this lead to differences in the embedding space or quality? Additionally, would there be inherent biases due to the different pretraining datasets used for these models?

**Methods And Evaluation Criteria:**

Their evaluation criteria is reasonable to me, and most of the experimental setups align with standard practices in the data selection field.

**Other Comments Or Suggestions:**

N/A

**Other Strengths And Weaknesses:**

I suggest investigating the domain shift before and after data augmentation. If a significant shift exists, should we consider training with the original data points as well? If there is no significant shift, would it be necessary to repeatedly update the embedding space during training?

**Questions For Authors:**

See my questions above.

**Relation To Broader Scientific Literature:**

The key contribution of this paper is highlighting the downside of data selection—namely, **the reduction of data diversity and its potential impact on model generalization**. To address this, they propose **an integrated framework** that combines data selection and data augmentation in a unified approach, which is novel to data selection literature.

**Theoretical Claims:**

There is no major parts for theoretical claims but i have walked through their problem setup. It looks correct to me.

---

> ### Author Rebuttal · Authors · 2025-03-31
>
> Dear Reviewer bQXA,
>
> We sincerely thank you for the careful review and insightful comments/questions. We appreciate your recognition of our work's strengths, e.g., novelty and empirical effectiveness.
>
> For the comments and questions, we provide our responses here:
>
> - **Q1: The impact of data selection on generalization.**
> - **A1:** Thank you for your insightful comment.
> To assess the impact of data selection on generalization, we have evaluated models trained on the selected data across various benchmark datasets (Tab. 1/3/4/5/6, Fig. 3,), which are commonly used to assess generalization performance. The results present a consistent trend across all baselines, as the selection ratio decreases, model test performance degrades, indicating reduced generalization. Despite this overall degradation, our method effectively alleviates this degradation by integrating data augmentation (DA) and selection.
>
> - **Q2: Clarification on the data used for training.**
> - **A2:** We would like to emphasize that only the augmented data is used for training. This follows common practice in many widely used DA and selection methods (e.g., TrivialAug, AutoAugment, InfoBatch, Moderate, etc.), where only the augmented version of selected data is used for training.
> Retaining the original and augmented versions would **double** the number of training data used for model forward and backward passes, thereby doubling the computation costs. This would **substantially undermine the training efficiency** that our method is designed to achieve.
>
>     While only augmented data is used for training, our augmentation strategy is carefully designed to preserve semantic consistency. Please refer to **A3** and **A4** for details.
>
> - **Q3: Semantic alignment before and after augmentation.**
> - **A3:** Thanks for the insightful comments. While our method does not explicitly verify each augmented data post-augmentation, our DA strategy is designed to prioritize semantic alignment.
> Specifically, each selected data is augmented using **only one operation** with its magnitude contained within a **moderate, predefined range** (Sec. A.1).
> In contrast, widely-used methods, such as AutoAug, Fast-AA, and RandAug, typically apply **2-3 operations per sample** with stronger transformation strengths, emphasizing more diverse training data rather than maintaining semantic consistency.
>
>     To further address your concern, we investigate the semantic alignment with a vanilla pretrained CLIP model using cosine similarity. Tab. B-1 shows that **the augmented data maintains a strong semantic consistency** with the original data.
>
>     While our current method does not include explicit post-augmentation verification to maintain high efficiency, we agree that incorporating a lightweight verification mechanism, without compromising efficiency, is a promising direction for future work, which is also a key challenge in DA research.
>
> **Table B-1**: Semantic alignment between original and augmented data points.
> ||C-10|C-100|
> |-|-|-|
> |Avg. Sim.|0.94|0.93|
> |Std.|0.06|0.05|
>
> - **Q4: Clarification on local structure after augmentation.**
> - **A4:** To further assess the local structural stability, we investigate changes in each sample's local nearest neighbors before and after augmentation. As shown in Tab. B-2, the proportion of altered neighbors is extremely low (<= 0.3%), validating that our DA introduces only minimal changes to local structure.
>
>     However, it is important to note that the feature space evolves continuously during training as the model updates. Thus, even if the augmented data remains semantically and structurally stable, it is **necessary to update the embedding space** to capture the model's current training state for dynamic data selection. This is also the core principle of dynamic data selection approaches.
> **Table B-2**: The ratio of changes in local nearest neighbors after augmentation.
> ||Change Ratio|
> |-|-|
> |C-10|0.2%|
> |C-100|0.3%|
> - **Q5: Clarification on the embedding generators.**
> - **A5:**  As clarified in Sec. 3.1, only the CLIP model is used to derive the semantic consistency distribution for filtering, which captures the intrinsic representativeness of training data. Meanwhile, as introduced in lines 207-213, to adapt CLIP to the target domain, we use a lightweight adapter to enable domain-specific knowledge transfer while preserving CLIP's strong alignment capabilities.
> On the other hand, the task model (e.g., ResNet or ViT series) is used to estimate the evolving density distribution during training. Thus, our framework minimizes the inherent biases.
>
> - **Q6: Details of augmentation operations.**
> - **A6:** For each sample, we apply **only one random augmentation operation** from Sec A.1. Following common practice in data augmentation (e.g., AutoAug, TrivialAug), we control the number of applied operations and the applied strength via a predefined, bounded magnitude range, ensuring consistency and avoiding overly distorted augmentation.

---

> > ### Comment · Reviewer_bQXA · 2025-04-02
> >
> > Thank you for your response. I have carefully reviewed your explanations.
> >
> > 1. Regarding the generalization results, I appreciate you pointing them out. I believe model generalization refers to the ability to handle unseen tasks or transfer to a new domain, rather than evaluating performance on noisy or corrupted samples. The experiments in the paper seem to be more related to assessing a model's robustness to incorrect samples.
> >
> > 2. Post-verification plays a crucial role in this work, and how it is controlled can significantly impact selection performance. Additionally, if semantic alignment remains strong, how does this influence/improve data selection?
> >
> > Thank you for your time and response.

---

> > > ### Author Response · Authors · 2025-04-03
> > >
> > > Dear Reviewer bQXA,
> > >
> > > Thanks for appreciating our work and giving the valuable comment.
> > >
> > > It is a point worth discussing. We would like to provide more discussion on your comments.
> > >
> > > - **Q7: Regarding the impact of data selection research on generalization.**
> > > - **A7:**
> > > We would like to emphasize that, beyond evaluating robustness to noisy scenarios, our experiments have also included a dedicated evaluation of the model's **generalization** on ImageNet-A, -O, -R, and -Hard, as shown in Table 4.
> > > These benchmarks are specifically constructed to evaluate generalization to **unseen domains and distribution shifts**.
> > > All these results present a similar trend: **as the selection ratio decreases, model generalization performance tends to degrade.**
> > >
> > > To further address your comments, we conducted an additional **cross-domain transfer learning** experiment in Table B-3, where models are pre-trained on ImageNet-1k and fine-tuned on CIFAR-10.
> > > These two datasets differ significantly in resolution, label granularity, and visual domain, making this a strong test of generalization across domains. The results show a similar pattern: lower selection ratios lead to reduced performance.
> > >
> > > Importantly, this trend has also been observed in many existing data selection literatures, e.g., Moderate, InfoBatch, DP, MoSo, etc. Thus, higher selection ratios are typically applied to ensure generalization.
> > >
> > >
> > > **Table B-3:** Evaluation of cross-domain generalization.
> > > |Selection Ratio (%)|10|20|30|50|60|70|80|90|
> > > |-|-|-|-|-|-|-|-|-|
> > > |Acc. (%)|85.2|85.6|85.9|86.5|86.8|86.9|87.2|87.7|
> > >
> > > We hope these clarifications and additions address your comment.
> > >
> > > - **Q8: Clarification on Post-Augmentation and Semantic Alignment.**
> > > - **A8:** Thank you for pointing this out. We address your comments in two parts.
> > >
> > > **1. Regarding Post-Augmentation:** Although our method does **NOT** include an explicit post-augmentation verification step to maintain high training efficiency, following the best practices in recent data augmentation (DA) research, we control the augmentation process by applying only one operation with its strength constrained within a predefined, moderate range.
> > > While not an explicit verification mechanism, this augmentation design effectively avoids overly distorted augmentation while preserving semantic consistency (as discussed in **A3** and **A4**).
> > >
> > > Moreover, this setting is **uniformly applied across all experiments**, and our results consistently show that both the model and selection performance remain stable and consistently superior, validating the effectiveness.
> > >
> > >
> > > **2. Regarding Benefits of Semantic Alignment:** Since our framework uses augmented data for training and constructing the embedding space, maintaining strong semantic alignment provides several key benefits:
> > >
> > > (1) Because semantic alignment is preserved, augmented low-density samples remain within low-density areas of the embedding space. This enables the model to effectively learn from underrepresented or insufficiently learned areas (as discussed in Fig. 2 and Sec. 3.2, paragraph 2).
> > >
> > > (2) Strong semantic alignment ensures that augmented samples remain meaningful and unambiguous. This is critical for accurately identifying **truly informative low-density samples**, rather than being misled by distorted or over-augmented inputs.
> > >
> > > (3) Prior DA research (TrivialAug, KeepAug, MADAug) has shown that preserving semantic structures enhances DA and model performance. In our framework, this contributes to improved model performance, and further **more accurate density estimation and sample selection in the dynamic data selection process** during training  (as discussed in Sec. 3.4, lines 206–219, page 4).
> > >
> > > Thank you again for your time and responses.

---

### Official Review · Reviewer_JbtQ · 2025-03-22

**Overall Recommendation:** 4

**Summary:**

This paper combines data augmentation and dynamic data selection. The main idea is to augment examples that the model is uncertain about, while filtering noisy examples using semantic consistency. The experimental results show that with only 50% training compute, equal performance can be achieved to training on the full dataset.

**Claims And Evidence:**

Yes, the claims are well supported.

**Essential References Not Discussed:**

Missing a reference to some key works exploring similar ideas:
- (Data-Efficient Augmentation) Liu, Tian Yu, and Baharan Mirzasoleiman. "Data-efficient augmentation for training neural networks." Advances in Neural Information Processing Systems 35 (2022): 5124-5136.
The paper above is perhaps the most relevant baseline for this paper as it too selects a subset of data to augment.
Other Relevant Work:
-  (Submodular Data Selection) Joshi, Siddharth, and Baharan Mirzasoleiman. "Data-efficient contrastive self-supervised learning: Most beneficial examples for supervised learning contribute the least." International conference on machine learning. PMLR, 2023.

**Experimental Designs Or Analyses:**

Looks reasonable.

**Methods And Evaluation Criteria:**

I think a key baseline is missing: (Data-Efficient Augmentation) Liu, Tian Yu, and Baharan Mirzasoleiman. "Data-efficient augmentation for training neural networks." Advances in Neural Information Processing Systems 35 (2022): 5124-5136.
This paper is perhaps the most relevant baseline for comparison as it similarly selects a subset of data for augmentation. It proposes a more theoretically rigorous approach to determine which examples are most useful for training. While uncertainty-based methods can sometimes prioritize difficult but unlearnable samples or even noisy samples, and semantic consistency might mitigate this issue, using model uncertainty as a heuristic for sample importance remains limited. The aforementioned paper instead leverages model gradients to identify the samples that contribute most significantly to learning, providing both theoretical guarantees and empirical validation of this approach's effectiveness.

**Other Comments Or Suggestions:**

I would highly recommend the authors to include the missing baseline and references. I think this would strengthen the paper significantly.

**Other Strengths And Weaknesses:**

N/A

**Questions For Authors:**

N/A

**Relation To Broader Scientific Literature:**

Discussed in essential references not discussed.

**Theoretical Claims:**

N/A.

---

> ### Author Rebuttal · Authors · 2025-03-31
>
> Dear Reviewer JbtQ,
>
> We sincerely thank you for the comments and constructive suggestions. We appreciate your recognition of our work's strengths, e.g., reasonable experimental designs and well-supported claims.
>
> For the comments, we provide our response as follows.
>
> - **Q1: Comparison with Data-Efficient Augmentation (DEA) [a].**
> - **A1**: Thank you for suggesting additional references and comparisons. We acknowledge the theoretical significance of Data-Efficient Augmentation, particularly its principled approach to rigorously determining which samples are most useful for training.
>
> In response to your suggestion, we compared our method with the suggested data-efficient augmentation using the same amount of training data and number of training iterations, across standard benchmarks (Table A-1), and under challenging noisy and corrupted scenarios (Table A-2). For ImageNet-1k, we utilized the reported results from [a].
>
> The results show that our method consistently achieves higher accuracy than DEA across selection ratios and datasets, achieving at least 3% higher accuracy on ImageNet-1k, and notable gains on CIFAR-10 and Tiny-ImageNet.
> Moreover, as shown in Table A-2, our method demonstrates stronger robustness to both noisy and corrupted conditions.
>
>
>
> **Table A-1**：Comparison with [a] on CIFAR-10 with ResNet-18 and Tiny-ImageNet and ImageNet-1k with ResNet-50.
> |||C-10|||T-IN|||IN1k||
> |-|-|-|-|-|-|-|-|-|-|
> |Selection Ratio (%)|30|50|70|30|50|70|10|30|50|
> |[a]|83.4|88.1|88.6|31.9|36.9|43.5|68.5|72.0|73.3|
> |Ours|**94.9**|**95.5**|**96.0**|**44.9**|**47.0**|**49.4**|**71.5**|**75.0**|**76.5**|
>
>
> **Table A-2**: Comparison with [a] under noisy and corrupted conditions using Tiny-ImageNet with ResNet-50.
> ||Noisy||Corrupted||
> |-|-|-|-|-|
> |Selection Ratio (%)|20|30|20|30|
> |[a]|25.8|28.2|31.8|31.9|
> |Ours|**35.9**|**39.6**|**39.1**|**42.0**|
>
> [a] Liu, Tian Yu, and Baharan Mirzasoleiman. "Data-efficient augmentation for training neural networks." NeruIPS 2022.
>
> - **Q2: Suggested References.**
>
> - **A2:** Thank you for suggesting more related works.
> Since the revised manuscript can not be updated at the current stage, we will include these references in Section 2.1 in the final version. Specifically,
>
>     - Sec 2.1, paragraph 3: add references "*The work (Liu \& Mirzasoleiman, 2022) proposes a theoretically rigorous approach to determine impactful data for training. SAS (Joshi \& Mirzasoleiman, 2023) improves data efficiency in SSL by proving and selecting the most beneficial data for contrastive training.*"
>
> We hope these additions and clarifications address your comments.

---

> > ### Comment · Reviewer_JbtQ · 2025-04-01
> >
> > Thanks for addressing my comments, I'd like to stick my rating to "accept" this paper!

---

> > > ### Author Response · Authors · 2025-04-02
> > >
> > > Dear reviewer JbtQ,
> > >
> > > We would like to express our sincere gratitude to reviewer JbtQ for acknowledging our work and providing insightful comments.
> > >  Thanks again for the time and effort in reviewing our work.

---

### Decision · Program_Chairs · 2025-05-01

**Decision:**

Accept (poster)

**Comment:**

This paper proposes a framework which combines dynamic data selection and data augmentation to improve training efficiency without sacrificing generalization. By selecting low-density, semantically consistent samples for augmentation, the method achieves strong empirical performance across several benchmarks and architectures.

Three reviewers recommended acceptance, as they found the motivation clear, the integration of ideas effective, and empirical performance solid, and ablations well-designed. However, one reviewer raised a concern about whether robustness to noise adequately supports generalization claims. The authors addressed this with additional evaluations on ImageNet-A/O/R/Hard and a cross-domain transfer task, which helped mitigate the concern from the reviewer, but was not strong enough to update the score. Yet, there was a consensus among the reviewers to accept the paper. One remaining limitation is that the method trains only on augmented samples, which may limit its applicability to generative models where preserving the original data distribution is more important.